# Systems Vaccinology for a Live Attenuated Tularemia Vaccine Reveals Unique Transcriptional Signatures That Predict Humoral and Cellular Immune Responses

**DOI:** 10.3390/vaccines8010004

**Published:** 2019-12-24

**Authors:** Muktha S. Natrajan, Nadine Rouphael, Lilin Lai, Dmitri Kazmin, Travis L. Jensen, David S. Weiss, Chris Ibegbu, Marcelo B. Sztein, William F. Hooper, Heather Hill, Evan J. Anderson, Robert Johnson, Patrick Sanz, Bali Pulendran, Johannes B. Goll, Mark J. Mulligan

**Affiliations:** 1Hope Clinic, Division of Infectious Diseases, Department of Medicine, Emory University School of Medicine, Decatur, GA 30030, USA; msnatrajan@gmail.com (M.S.N.); nroupha@emory.edu (N.R.); lilin.lai@nyulangone.org (L.L.); dweiss8@emory.edu (D.S.W.); 2Emory Vaccine Center, Emory University, Atlanta, GA 30322, USA; dkazmin@emory.edu (D.K.); cibegbu@emory.edu (C.I.); bpulend@stanford.edu (B.P.); 3The EMMES Corporation, Rockville, MD 20850, USA; tjensen@emmes.com (T.L.J.); whooper@emmes.com (W.F.H.); hhill@emmes.com (H.H.); 4Center for Vaccine Development and Global Health, University of Maryland School of Medicine, Baltimore, MD 21201, USA; msztein@som.umaryland.edu; 5Departments of Medicine and Pediatrics, Emory University School of Medicine, Atlanta, GA 30322, USA; evanderson@emory.edu; 6Biomedical Advanced Research and Development Authority, ASPR, Department of Health and Human Services, Washington, DC 20201, USA; Robert.Johnson@hhs.gov; 7Division of Microbiology and Infectious Diseases, National Institute of Allergy and Infectious Diseases, National Institutes of Health, Rockville, MD 20892, USA; patrick.sanz@nih.gov; 8Institute for Immunity, Transplantation and Infection, Department of Pathology, Department of Microbiology and Immunology, Stanford University, Stanford, CA 94304, USA; 9Division of Infectious Diseases and Immunology, Department of Medicine, and New York University (NYU) Langone Vaccine Center, NYU School of Medicine, New York, NY 10016, USA

**Keywords:** tularemia, transcriptomics, immune signaling, comparative vaccines, live attenuated vaccines

## Abstract

*Background:* Tularemia is a potential biological weapon due to its high infectivity and ease of dissemination. This study aimed to characterize the innate and adaptive responses induced by two different lots of a live attenuated tularemia vaccine and compare them to other well-characterized viral vaccine immune responses. *Methods:* Microarray analyses were performed on human peripheral blood mononuclear cells (PBMCs) to determine changes in transcriptional activity that correlated with changes detected by cellular phenotyping, cytokine signaling, and serological assays. Transcriptional profiles after tularemia vaccination were compared with yellow fever [YF-17D], inactivated [TIV], and live attenuated [LAIV] influenza. *Results:* Tularemia vaccine lots produced strong innate immune responses by Day 2 after vaccination, with an increase in monocytes, NK cells, and cytokine signaling. T cell responses peaked at Day 14. Changes in gene expression, including upregulation of *STAT1*, *GBP1*, and *IFIT2*, predicted tularemia-specific antibody responses. Changes in *CCL20* expression positively correlated with peak CD8+ T cell responses, but negatively correlated with peak CD4+ T cell activation. Tularemia vaccines elicited gene expression signatures similar to other replicating vaccines, inducing early upregulation of interferon-inducible genes. *Conclusions:* A systems vaccinology approach identified that tularemia vaccines induce a strong innate immune response early after vaccination, similar to the response seen after well-studied viral vaccines, and produce unique transcriptional signatures that are strongly correlated to the induction of T cell and antibody responses.

## 1. Introduction

Tularemia is caused by *Francisella tularensis*, a gram-negative bacterium, and is classified as a Tier 1 select agent due to its high virulence and potential use in bioterrorism, with mortality rates of ~30% if untreated [1,2]. After infection with *F. tularensis,* initial immune responses are nonspecific and include macrophage activation followed by later development of cellular and humoral immunity [3,4]. Animal studies have previously shown that B and T cell responses are important in the induction and regulation of an effective immune response to live tularemia vaccines [5,6]; specifically, maintaining either CD4+ or CD8+ T cells in mice appeared to be essential for survival. Animals challenged with virulent *Francisella* require both T cell subsets for survival [7,8], and in vitro studies of human cells suggest that CD8+ T cell proliferation and cell survival depend on CD4+ proliferation [9]. However, the natural course and efficacy of acquired immunity is not well studied in large human cohorts.

*F. tularensis* live vaccine strains (LVSs) have been delivered by scarification since the 1950s and have been shown to be protective against tularemia [10]. Two investigational lots of a vaccine have been developed against tularemia, the newer Dynport Vaccine Company live vaccine strain (DVC-LVS), derived from Lot 4 of the U.S. Army Medical Research Institute of Infectious Diseases live vaccine (USAMRIID-LVS), which was produced in the 1960s. Both lots were previously found to be safe and immunogenic [11,12] and were derived from 100% blue colonies of bacteria harvested from blood agar plates [13], with the main difference being that the DVC-LVS lot used updated Good Manufacturing Practices in the early 2000s. Other studies in a small cohort of six subjects have shown that tularemia LVS can induce similar, efficient innate cell responses in different subjects [14,15], but these studies did not produce results of serum cytokine responses or correlate gene expression patterns and potential biomarkers to antibody or cellular responses.

A large study of human transcriptional and innate/adaptive cell signatures activated by tularemia vaccines has not been performed, and the transcriptomic responses to tularemia LVS have not been compared to other effective vaccines. The yellow fever vaccine (YF-17D) is a live, attenuated viral vaccine that results in a potent immune response, including strong memory CD8+ T cell responses and the amplification of pathways that regulate virus sensing and type 1 interferon production [16,17]. The trivalent inactivated (TIV) and live-attenuated (LAIV) influenza vaccines are used in the prevention of influenza, and peripheral blood mononuclear cells (PBMC) gene expression and immune cell signatures have shown unique mechanisms of immunogenicity, with TIV producing predominant B cell responses and LAIV producing predominant T cell responses [18]. Changes in gene expression profiles have been shown to be predictive of antibody responses in YF-17D, TIV, and LAIV [19]. Using a similar approach following tularemia vaccination may reveal predictive biomarkers of a positive immune response and lead to improved *F. tularensis*, or perhaps other, live-attenuated bacterial vaccine candidates.

In this study, we used a systems biology to provide a deeper understanding of the immunologic responses to the tularemia vaccine [20]. We assessed changes in immune cell populations, cytokine concentrations, and gene expression in PBMCs after tularemia vaccination and identified molecular signatures that predicted the magnitude of lymphocyte and antibody responses. We compared our results to two other live, attenuated vaccines (YF-17D, LAIV) and one inactivated vaccine (TIV) in order to identify correlates of protection against a live bacterial vaccine compared to other commonly used viral vaccines.

## 2. Materials and Methods

### 2.1. Vaccine Used and Specimen Collection

Two live attenuated tularemia vaccine lots were administered by scarification with a bifurcated needle as a single dose: the older USAMRIID-LVS produced by the National Drug Biologic Research Company (Swiftwater, PA, USA) and the newer DVC-LVS manufactured by DVC (Frederick, MD, USA), as described [12]. Blood samples were obtained from 42 healthy male and female subjects aged 18 to 45 years old, enrolled in DMID 08-0006 (NCT01150695) at Emory University. The BD Vacutainer^®^ CPT™ Cell Preparation Tubes with Sodium Citrate were used to collect PBMC and plasma samples from subjects. Laboratory assessments included gene expression based on PBMC RNA using microarrays, serum cytokine/chemokine concentration measurements, antibody responses, and immune cell phenotyping (DC/monocytes/natural killer/lymphocytes). The study was reviewed by the Emory Institutional Review Board (IRB) and informed consent was obtained from all participants.

### 2.2. Microarray Experiments

Affymetrix High-throughput (HT) HG-U133 PM Array GeneChips were run in a 96-array plate configuration (96 samples per HT array). Each individual array contained 54,715 probe sets and 536,460 perfect match (PM) probes (on average 9.8 probes per probe set). Three 96 HT array plates were run producing data for 205 samples from five time points (baseline prior to vaccination on Day 0 and Days 1, 2, 7, and 14 after vaccination). The RMA algorithm [21] was used to obtain background-corrected and quantile-normalized probe-set intensities. The ComBat algorithm [22] using the parametric empirical Bayes option was applied to adjust the probe-level data for observed batch effects for HT array plates. Probe sets with a sample coefficient of variation less than or equal that of the 25% quantile were filtered out. For comparative analysis, RMA-normalized expression data (Days 0, 3, 7) were obtained for YF-17D [17], TIV, and LAIV [18]. To align different Affymetrix GeneChip types (HG-U133 PM for tularemia vaccine and HG133 Plus 2 for the others), the mean log_2_ fold change across gene sets that mapped to the same gene symbol (based on Affymetrix release 34 probe set annotations) was used for comparative analysis.

### 2.3. Cellular Phenotyping

Immune cells were phenotyped by using flow cytometry at baseline and on Days 1, 2, 7, 14, 28, and 56 as previously described [23] using an LSRII cell analyzer (BD Biosciences, Franklin Lakes, NJ, USA) and FlowJo Software (TreeStar, Ashland, OR, USA). All samples were gated to exclude dead cells, debris, and doublets. Dendritic cells (CD3-CD14-CD16-CD19-CD20-CD56-HLA-DR+) were defined as myeloid (mDCs, CD11c+) or plasmacytoid (pDCs, CD123+) and further classified by BDCA1, CCR7, CD86, and CD11b expression. Monocytes (CD19-CD3-HLA-DR+) were subdivided by CD14/CD16 expression and further with CD86 and CCR5. Lymphocytes (CD14−) were divided to T cells (CD3+), B cells (CD19+), and natural killer cells (CD3-CD19-CD56dimCD16+) and further categorized based on CD4 (CD4+ and CD4− for the Leukocyte Panel) and CD8 (for T cell phenotyping), CCR7, CD86, CD11b, HLA-DR, CCR5, CD38, and CD69. Activated T cells were defined by CD38 and HLA-DR co-expression or intracellular co-expression of Ki67 and Bcl-2. CD4 responders were defined at Days 4, 14, or 28 by percentage of CD3+CD4+ CD38+HLA-DR+ effector T cells at least 2 standard deviations (SD) above the mean of all baseline samples. CD8+ T cell responders were defined at Days 7, 14, or 28 by percentage of CD3+CD8+ CD38+HLA-DR+ effector T cells at least 3 SD above the mean of all baseline samples.

### 2.4. Plasma Cytokine Assay

Human plasma cytokines were analyzed at Days 0, 1, 2, 7, and 14 using the Bio-Plex Pro Human Cytokine 27-plex Assay (Bio-Rad, Hercules, CA, USA) according to the manufacturer’s instructions.

### 2.5. Microagglutination Antibody Assay

Antibody responses were measured by a tularemia-specific microagglutination assay performed as previously described at Days 0, 14, and 28 [12]. Antibody responders were defined by an increase in tularemia-specific log-transformed microagglutination titers >3 SD above the mean of all log-transformed baseline values at Days 14 or 28. See Appendix A for additional details.

### 2.6. Statistical Analyses

The cluster R package (Version 1.14.4, R, Boston, MA, USA) was used for k-means clustering. A two-sided paired *t*-test was applied to identify differentially expressed genes (DEGs) relative to pre-vaccination while a two-sided Welch’s *t*-test was used to identify DEGs between vaccine groups based on mean log_2_ fold change difference. The false discovery rate (FDR), which controls the false positive rate among DEGs, was calculated using the *q*-value R package [24] (Version 1.34, R, Boston, MA, USA). Genes with a *q*-value ≤ 0.05 and a fold change of ≥1.5-fold in either direction were deemed DEGs. To identify robust clusters of co-expressed genes based on log_2_ fold change over time, multiscale bootstrapping was carried out using the pvclust R package (Version 1.2.2, R, Boston, MA, USA). Gene set enrichment analysis was performed using the Java implementation (Version 2.1.0, Java, Redwood City, CA, USA) of the GSEA algorithm (GSEA Preranked method) to detect enriched gene sets [25]. The *mixOmics* (v5.0–3), and *glmnet* (v2.0–2), R packages (R, Boston, MA, USA) were used for regularized canonical and logistic regression analysis to identify gene responses (based on log_2_ fold change) that correlated with changes in cytokines/antibody or predicted a positive serological and T-cell immune response, respectively. In both cases, leave-one-out cross-validation was used to select optimal models. As there was no a priori knowledge about the correlates of protection for tularemia, for logistic regression analysis, subjects that achieved a response that exceeded the mean response of all pre-vaccination samples by 3 SD (percent activated CD8+ T cells and microagglutination titer) or 2 SD (percent activated CD4+ T cells) were classified as positives. A less restrictive cut off for CD4+ T cells was chosen to have at least 10 positive responders. See Appendix A for additional detail on the methods.

## 3. Results

### 3.1. Tularemia Vaccination Induced Peak Innate Responses at Day 2 and Peak Adaptive Responses at Day 14

Cellular phenotyping and plasma cytokine/chemokine assays were performed to characterize the immune response after tularemia vaccination at Days 1, 2, 7, and 14 relative to baseline (Figure 1). For both vaccine lots, monocyte (CD16+) and natural killer cell (CD56dimCD16-CD69high) activation increased at Day 1, reaching peak levels by Day 2, and decreased to pre-vaccination levels by Day 7 (Figure 1A,B). In contrast, T cell (CD4+HLA-DR+ and CD4−HLA-DR-) activation reached peak responses at Day 14 (Figure 1A,B). The DVC-LVS lot produced a stronger CD4−HLA-DR+ T cell response at Day 7 (Figure 1A), and the USAMRIID lot showed a more pronounced monocyte (CD16+) response at Days 1, 2, and 7 (Figure 1B). Plasma markers of early inflammation and innate signaling increased at Days 1 and 2, including peak increases in IFN-γ and human interferon gamma-induced protein 10 (IP-10). In addition, both vaccine lots produced elevated IL1RA and IL4 at Day 2 (Figure 1C,D). Overall, the change in the platelet-derived growth factor (PDGF) showed the strongest increase in the DVC-LVS group at Days 1 and 2 (Figure 1C) and was not observed in USAMRIID recipients (Figure 1D). For the DVC-LVS group, the strongest cytokine responses (PDGF, IL-1RA, IFN-γ) that peaked at Day 2 had lower than pre-vaccination concentrations at Day 7 (Figure 1C). Descriptive statistics for changes in cell types and cytokines/chemokines after administration of each vaccine can be found in the Appendix A).

### 3.2. DVC-LVS and USAMRIID-LVS Elicited Similar Gene Expression Profiles

To compare post-vaccination transcriptomic responses between the two tularemia vaccine lots, we assessed differentially expressed genes (DEGs) within and between groups as well as log_2_ fold change profiles of the union of DEGs using k-means clustering (using k = 2). For both groups, up- and down-regulation of genes increased between Days 1, 2, and 7 and remained high at Day 14 (Figure 2A). Comparing pre-versus post-vaccination resulted in 43 DEGs in the DVC-LVS group and 107 DEGs in the USAMRIID-LVS group (Figure 2A in red). The directionality of these responses (up/down-regulation) was the same (Appendix A). For 41% of DEGs, fold change magnitude exceeded the 1.5-fold change cut off for both lots. K-means clustering of log_2_ fold changes using the union of DEGs did not show a group effect for any post-vaccination day (Figure 2B). When comparing log_2_ fold changes directly between the two groups using a two-sided Welch’s *t*-test, no significant differences were observed. As comparisons did not reveal substantial differences, gene expression profiles from the two tularemia groups were combined for subsequent analyses. With this combined dataset, we identified 421 unique DEGs (Appendix A).

### 3.3. Gene Clustering Analysis Revealed Co-Expressed Genes That Were More Highly Upregulated at Day 7 in the DVC-LVS Group and Correlated with Increased Antibody Responses at Day 14

The visualization of time trends of robust clusters of DEGs revealed that most co-expressed DEGs had similar response profiles for both vaccine lot groups over time (Appendix A). The most noticeable difference was observed for a cluster of immunoglobulin genes. While mean cluster responses were increased by 46% at Day 7 for the DVC-LVS group, responses reached comparable levels by Day 14 (Figure 2C). In line with our previous study results [12], subjects in the DVC-LVS group achieved higher GMTs at Day 14 (*p* = 0.014) (Figure 2C). To investigate the relationship between changes in gene expression and antibody responses, we assessed the correlation between mean cluster fold changes at Day 7 with antibody titers at Day 14 (Figure 2D). The observed positive correlation (r = 0.48) was statistically significant (*p*-value = 0.003).

### 3.4. Comparative Analysis of Gene Expression Patterns Following Tularemia, Yellow Fever, and Influenza Vaccination

To evaluate gene expression signatures induced following tularemia vaccination in the context of other vaccines, we compared our tularemia results (n = 38) at Days 2 and 7 with those observed at Days 3 and 7 for YF-17D (n = 25) [17], TIV (n = 26), and LAIV (n = 28) from different studies [18].

Vaccines elicited distinct responses with an overall higher number of DEGs at Day 7 compared to Day 2/3, irrespective of vaccine, with 259 and 97 total DEGs, respectively (Figure 3, Appendix A). Most DEGs at Day 2/3 were DE at Day 7 (89 of 97). Except for a subset of YF-17D subjects, heatmap analysis of DEGs at Day 2/3 did not show distinct vaccine group signals. Most subjects showed an upregulation of a cluster of 52 genes, indicating a generic immune system response (Figure 3A, purple bracket). Upregulation of these genes, many of which are known to play a role in interferon signaling, was most pronounced for subsets of tularemia and YF-17D recipients. At Day 7, there was a more appreciable difference in the overall transcriptional response (Figure 3B); YF-17D strongly upregulated a cluster of 77, mainly interferon-inducible genes at Day 7 (Figure 3B, purple bracket), while TIV induced a more potent increase for a cluster of 28, mostly immunoglobulin, genes (Figure 3B, orange bracket).

To further assess the temporal aspect of these responses, we visualized time trends for each cluster (Appendix A). Results showed that peak responses in interferon-related clusters were observed at Days 2/3 after tularemia and influenza vaccination while YF-17D was observed at Day 7 (Appendix A).

At Day 2, 23 genes significantly differed in their response following tularemia vaccination compared to one or more of the other vaccines (Appendix A). Tularemia vaccine induced higher guanylate binding protein (GBP) responses at Day 2, with upregulation of genes encoding for *GBP1P1*, *GBP5*, *GBP1*, and *GBP4* compared to the other vaccines at Day 3 (Figure 4A, blue bracket, Appendix A). By Day 7, responses for 160 genes differed (Figure 4B). This included 42–44 genes that showed differential responses following tularemia vs. LAIV/TIV (Appendix A). In contrast, 128 genes differed between tularemia and YF-17D (Appendix A). Seventeen genes had strongly elevated responses after tularemia vaccination compared to at least one other vaccine at Day 7 (Figure 4B, purple bracket). Approximately 50 genes, many of which are known to play a role in interferon signaling, were increased after tularemia vaccination compared to TIV/LAIV, but had relatively decreased responses compared to YF-17D vaccination (Figure 4B, orange bracket).

To further characterize DEGs, we carried out pathway enrichment analysis using Reactome [26]. The analysis revealed that Day 2 tularemia responses were functionally most similar to YF-17D at Day 7 as well as YF-17D at Day 3 (Figure 5). This similarity was primarily driven by enrichment of cytokine and interferon signaling (Figure 5, blue bracket). In contrast, the Day 7 DVC/USAMRIID-LVS response was most similar to LAIV responses at Day 7. Pathway enrichment results also showed that only live vaccination resulted in a significant induction of IFN-α/β signaling-related responses at Day 2/3, while such responses were undetectable with the inactivated TIV (Figure 5).

### 3.5. Gene Expression Signatures Predicted Antibody Responses

Overall, 28 subjects were classified as achieving a positive response based on their peak microagglutination titer, while 11 subjects were classified as having a negative response. Of these 39 total subjects, six subjects were positive responders for antibodies, CD4+ T cells, and CD8+ T cells against tularemia vaccines. Of the remaining positive antibody responders (n = 22), six were also CD4+ T cell responders and 12 were also CD8+ T cell responders. There were no subjects that were only CD4+ T cell responders, and five subjects that were only CD8+ T cell responders.

To identify gene expression responses following tularemia vaccination that best predicted serological responses, we applied regularized logistic regression analysis in combination with 5-fold cross-validation (Appendix A). At Day 2, 19 genes predicted a response in the logistic regression analysis with a mean cross-validated misclassification error of 26% (Appendix A). Of these, six (32%) were known to be involved in cytokine signaling. For 4 known interferon signaling-related genes, *STAT1*, *IFIT1*, *IFIT2*, and *GBP1*, a relative increase in gene expression at Day 2 increased the odds for a positive microagglutination response (Figure 6A). Among predictive genes, *IFIT2* and *GPR84* had the strongest positive and negative correlations with peak microagglutination titer, respectively (Figure 6B). At Day 7, 31 genes were selected, of which 7 (23%) were known to be involved in signal transduction (Figure 6A). Among predictive Day 7 genes, *TYMS* and *MIR155/MIR155H* had the strongest positive and negative correlations with peak microagglutination titer, respectively (Figure 6B).

Gene expression signatures predicted peak changes in activated CD4+ T-cell or CD8+ T-cells.

Next, we applied regularized logistic regression analysis to assess associations with a positive (activated) CD8+ (CD3+CD8+CD38+HLA-DR+) and CD4+ (CD3+CD4+CD38+HLA-DR+) T-cell phenotypes post-vaccination.

For CD8+ T cells, 23 subjects were considered to be positive responders. The best model for CD8+ T cells was identified for Day 1 with a mean cross-validated misclassification error of 22%, and 22 genes were selected. The highest impact on a positive CD8+ T-cell activation response was observed for increased expression of *CCL20* with higher fold changes for positive versus negative responders. Increased expression of Fc-gamma receptor genes *FCGR1A*, *FCGR1B*, and *FCGR1C* involved in antigen processing and cross-presentation reduced the odds for achieving a higher level of CD8+ T-cell activation (Figure 7A).

For CD4+ T cells, only 12 subjects were considered positive responders, and the best model was identified for Day 2 with a mean cross-validated misclassification error of 22%, and 11 predictive genes were selected (Appendix A). Increased expression levels for *CTH* had a negative impact on the odds of a positive CD4+ T cell activation response. While the responder group had 61% decreased levels at Day 2, the non-responder group had an average reduction of 24%. At Day 14, expression of *LRRC32*, a gene known to negatively regulate activated T cell proliferation, negatively impacted the odds for CD4+ T cell activation. The Day 1 CD4+ T cell model identified 7 predictive genes (Figure 7A). The model showed that the strongest impact for changes in *CCL20* occurred at Day 1. However, in contrast to the CD8+ T cell model, the relationship reversed. To identify patterns that explain associations between changes in gene expression and peak changes in CD3+CD4+HLA-DR+CD38+, CD3+ CD8+ HLA-DR+ CD38+, and CD3+CD8+BCL2+Ki67+ T-cells, we used regularized canonical correlation analysis (Appendix A). The best model was identified at Day 1. At Day 1, *CCL20* fold changes negatively correlated with peak activated CD4+ T-cells but positively correlated with activated CD8+ fold changes. Similar correlation patterns were observed for *NR4A3, OLR1, IL1A, ZNF331, PHACTR1, RASGEF1B,* and *PRMT10*.

We further investigated the relationship between the *CCL20* and peak CD4+ and CD8+ T cell response using scatterplots and Pearson correlation (Figure 7B). Results confirmed the inverse relationship identified in the logistic regression and canonical correlation models. The same pattern was observed for *NR4A3* at Day 1 (Figure 7C), which was also selected as a predictor in the CD4+ T cell logistic regression model (Figure 7A).

## 4. Discussion

Immune correlates of effective vaccination can provide helpful endpoints for clinical trials in the development of novel vaccines or in optimizing existing vaccines. Systems biology approaches provide insights into molecular underpinnings of such correlates and help further our understanding of molecular mechanisms that lead to a successful immune response. Here, we utilized a systems vaccinology approach to better understand molecular responses following *F. tularensis* vaccination and compare them to well-characterized viral vaccine responses. Using this approach, we identified gene signatures that were activated following vaccination, that were predictive of later effective innate and/or adaptive immune responses, and that were unique or shared compared to viral vaccines.

When we assessed differences in tularemia vaccines lots, we found that many of the transcriptomic responses were similar in direction and magnitude between lots, with a few differences that were worth noting. Co-regulated genes that indicated increased B-cell activation for the DVC-LVS group at Day 7 included *TNFRSF17*, which encodes a receptor expressed on mature B cells, and several immunoglobulin genes (*IGLJ2*, *IGLJ3,* and *IGLV2-14)*. However, by Day 14, the activation of this cluster was similar between the two groups. Interestingly, this lag in the transcriptional response for USAMRIID-LVS was also reflected in higher microagglutination titers at Day 14 compared to Day 28 [12]. At the same time, we observed an earlier increase in the activation of CD4+ T-cells for the DVC-LVS group, which reached near peak levels by Day 7 in contrast to Day 14 for the USAMRIID-LVS group. A difference in characteristics or age of the two vaccine lots (e.g., age 50+ years for USAMRIID-LVS versus 10 years for DVC-LVS) may have impacted in vivo growth curves of each one, and thus the timing of activation of cell signaling pathways. Additionally, the changes in the handling and development of the two vaccine lots may have contributed to the differences in certain responses, such as the increased monocyte response observed in the USAMRIID-LVS lot.

Comparisons of the tularemia vaccine with three different vaccines on gene expression levels revealed multiple shared and several unique DEGs. Datasets for these three vaccines were very similar in study design and array technologies to the tularemia vaccine gene analyses, so they were used to put findings into a broader context of immune responses to different types of vaccines (viral versus bacterial, live versus inactivated with different routes of administration). The tularemia vaccination induced a similar innate cell and transcriptional response relative to the YF-17D and influenza vaccines [18,27,28]. Results at the pathway level showed that Day 2 tularemia vaccine responses were most similar to Day 7 YF-17D. This agreement was primarily driven by cytokine and interferon signaling-related genes indicating that tularemia live attenuated vaccination triggered these signaling pathways earlier, albeit not as strongly as YF-17D at Day 7. By Day 7, interferon signaling was no longer enriched in PBMCs from tularemia vaccine recipients. Only live attenuated vaccines significantly induced interferon a/ß signaling-related responses at Day 2/3, suggesting early innate responses that assist with strong and long-lasting immunity expected after live vaccines. A previous study comparing LAIV and TIV responses also showed that subjects receiving LAIV demonstrated higher type 1 interferon-inducible gene signature scores [29].

Tularemia vaccination also induced changes in 23 early signatures that were not seen after other vaccines, including greater induction of members of the GBP family that are known to be induced by IFN-γ [30] and have been linked with control of *Francisella* infection in mice [31]. Although several interferon-inducible genes were associated with YF-17D, LAIV, and TIV, upregulation of GBPs was unique to tularemia. *GBP1* enhancement also predicted a higher peak antibody response, suggesting that activation of these genes plays a role in the magnitude of the humoral immune responses after tularemia vaccination. Three other genes correlated with antibody responses at this timepoint and are known to be induced by IFN-γ signaling: *STAT1* (transcription factor for interferon inducible genes), *IFIT1* and *IFIT2* (proteins that inhibit expression of viral messenger RNA [32]). *STAT1* also predicts delayed antibody responses in other vaccine studies [33,34].

T-cells were strongly upregulated at Day 14 in our study as well as in other vaccine studies [35,36]. The upregulation of activated CD8+ T-cells after tularemia vaccination strongly correlated with the expression of *CCL20* at Day 1. *CCL20* gene expression, known to be strongly chemotactic for lymphocytes [37], can be induced by microbial factors such as LPS, and inflammatory cytokines, and is downregulated by IL-10. *NR4A3* and *CCL20* have also been shown to positively modulate regulatory T-cells [38,39]. CCL20 is a key chemokine and potential biomarker that indicates subsequent T cell responses to the DVC/USAMRIID LVS vaccine. Thus, it is likely that complex reciprocal CCL20, CCR6 (the receptor for CCL20), and other gene co-regulation [40] led to our observation of differential correlations for activated T cells and CCL20. Other studies have seen that production of CCL20 by myeloid dendritic cells was crucial to priming of CD8+ T cells but not CD4+ T cells [41] and that NR4A2, a close relative of NR4A3, activity is crucial in Treg function and suppression of CD4+ T cells [42]. With live bacterial vaccine/infection, functional antagonistic co-regulation of the observed response by Th17 and Treg cells is likely to occur—an area for future investigation. Both DVC and USAMRIID-LVS also induced production of the cytokines IFN-γ and IP-10, which are known to be regulators of activated T lymphocytes and have been suggested as potential biomarkers for other bacterial infections such as *M. tuberculosis* [43].

Characterizations of memory B cell and plasmablast responses as well as validation of the correlation between gene responses (such as IFIT2 with microagglutination titers) were not performed due to limited cell numbers, but further studies of these responses will assist in understanding the importance of protective B cell responses seen in mouse studies [44]. A limitation of this study was the temporal difference in vaccine administration, with tularemia analyzed two days post vaccination and others analyzed three days post vaccination. This early time window may be critical in the innate immune response after vaccination and changes within that window could confound comparisons. This study was also performed using microarray technology that has largely been replaced by RNASeq. RNASeq has been shown to outperform microarrays due to its wider dynamic range [45,46,47]. However, these methods have been shown to be comparable in their prediction of clinical endpoints [48]. We ran RNASeq for eight subjects and found that the mean Pearson correlation between log_2_ fold changes of DEGs was 0.81, 0.83, 0.80, and 0.82 for Days 1, 2, 7, and 14, respectively, indicating a high correlation between DEG fold changes for microarray and RNASeq. In future studies, transcriptional responses to tularemia vaccines can be compared to datasets for other live bacterial vaccines, such as the BCG vaccine, to assist with determining whether the gene signatures and differences noted here are specific to tularemia or generalizable to differences seen between bacterial and viral vaccines.

## 5. Conclusions

These results indicate that after tularemia vaccination, early induction of innate cells, and interferon/chemokine signaling might play a key role in the magnitude of antibody and T cell responses. They also suggest that these innate signaling pathway results might be useful as early predictors (biomarkers) of subsequent protective antibody and cellular effector responses following *F. tularensis* vaccination (e.g., during a rapid response to a bioterror threat). Towards that end, this study revealed several targets that can be exploited further including GBP1, IFIT2, STAT1, and CCL20. It would be beneficial to further characterize the systems biology/immune response to tularemia vaccination by employing multiple additional ‘omics technologies’, such as proteomics, metabolomics, and lipidomics. Such an approach could be beneficial in advancing hypotheses related to innate and adaptive immunity following bacterial vaccination that could address opportunities for novel adjuvants or therapies for bacterial infections with live vaccines, such as *F. tularensis* and Tuberculosis.

## Figures and Tables

**Figure 1 vaccines-08-00004-f001:**
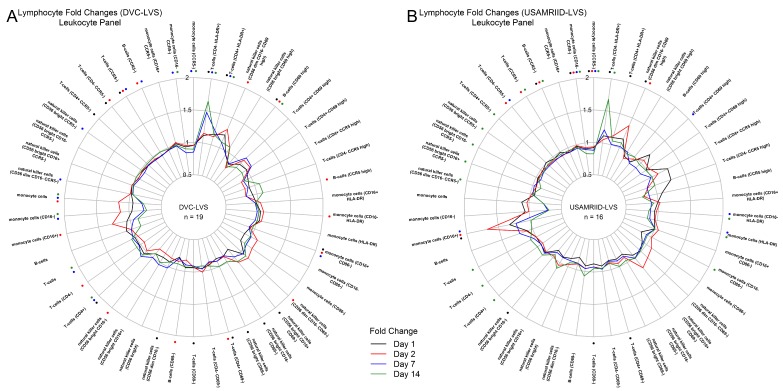
Immune cell and cytokine profiles following tularemia vaccination: innate immune responses at day 1 and 2 and adaptive responses at days 7 and 14. (**A**,**B**) Radar plots of median fold change in percent cells (FACS phenotyping) for each vaccine lot. Monocytes and natural killer cells are strongly upregulated at days 1 and 2. CD4+ T cell numbers increased at day 7 and were still increasing at day 14. The median responses in percent cells for each day are displayed. Asterisks color-coded by post-vaccination day indicate statistically significant changes compared to pre-vaccination (Wilcoxon signed-rank test *p*-value < 0.05). (**C**,**D**) A 27-plex Luminex was used to measure the concentration (pg/mL) of cytokines post-vaccination. Radar plots of median baseline fold change in cytokine/chemokine concentration are presented for each vaccine lot. Significant changes are marked with asterisks.

**Figure 2 vaccines-08-00004-f002:**
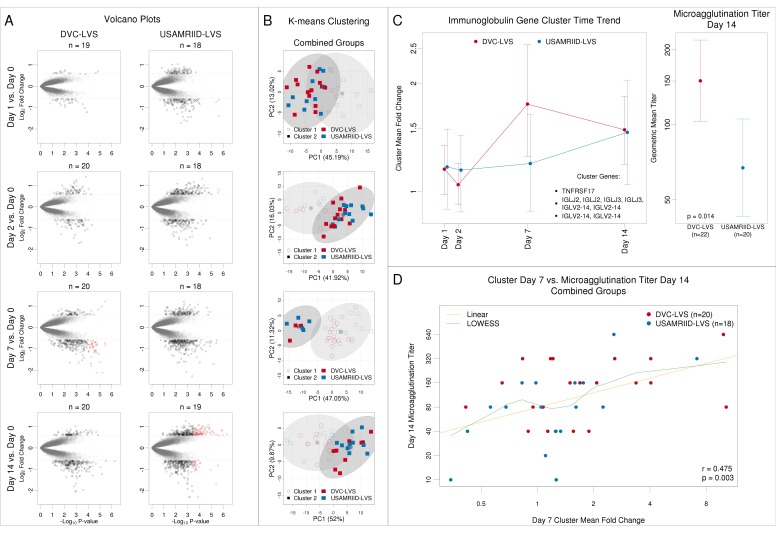
DVC-LVS and USAMRIID-LVS tularemia vaccine lots induce similar transcriptomic response profiles. (**A**) Volcano plots for the DVC-LVS (column 1) or USAMRIID-LVS (column 2) group contrasting the log_10_
*p*-value by log_2_ fold change from pre-vaccination (red = DEGs, FDR *p*-value ≤ 0.05; black = genes with ≥1.5-fold change; grey = genes with <1.5-fold change). 43 genes in the DVC-LVS and 107 in the USAMRIID-LVS were found to be differentially expressed for one but not the other vaccine lot but the directionality was similar. (**B**) K-means clustering analysis of log_2_-fold changes of genes that were differentially expressed in either vaccine group for any post-vaccination day showed clustering of both tularemia vaccine lots together. K-means clustering (colored squares) is overlaid on principal component biplot clusters (empty circles). (**C**) Fold changes for co-expressed genes that were significantly differentially regulated in the DVC-LVS group at day 7 are compared to USAMRIID at Day 7 (46% increase in DVC group). All genes had similar regulation at Days 1, 2, and 14. On the right panel, microagglutination antibody assay titers with associated 95% confidence intervals at Day 14 for the two groups (*p* = 0.014). (**D**) Scatterplot of mean cluster fold change in genes at Day 7 versus log_2_ microagglutination antibody assay titer at Day 14 (green line: locally weighted scatterplot smoothing fit; yellow line: linear fit).

**Figure 3 vaccines-08-00004-f003:**
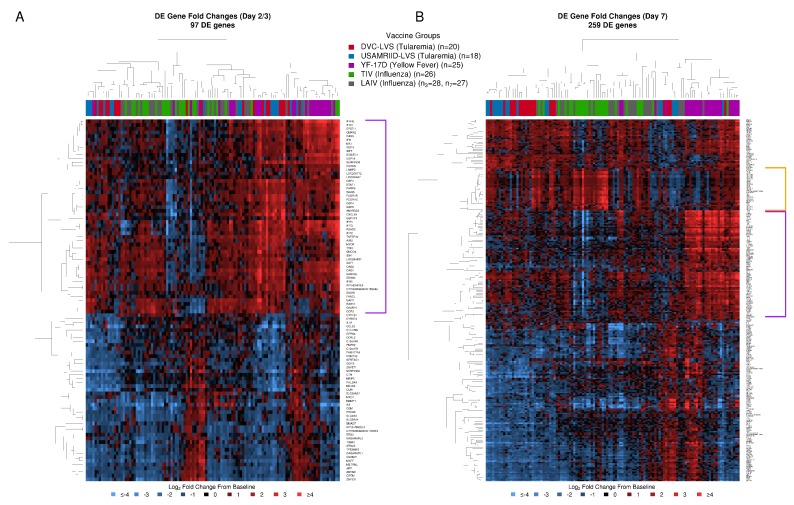
Transcriptomic profiles display specific differential gene expression after tularemia, yellow fever, and influenza vaccines at day 7 post vaccination. (**A**) 97 differentially expressed genes were identified at Day 2 post-tularemia vaccination, Day 3 following yellow fever (YF-17D), or Day 3 post-influenza immunization (LAIV and TIV) vaccination, with 50 cytokine-signaling related genes being upregulated in most subjects irrespective of vaccine type (highlighted in purple). (**B**) At Day 7, 259 differentially expressed genes were identified following yellow fever, influenza, or tularemia vaccination. The YF-17D vaccine induced a strong upregulation of interferon-inducible genes (highlighted in purple) while TIV induced plasmablast responses characterizes by an up-regulation of several immunoglobulin genes (highlighted in orange).

**Figure 4 vaccines-08-00004-f004:**
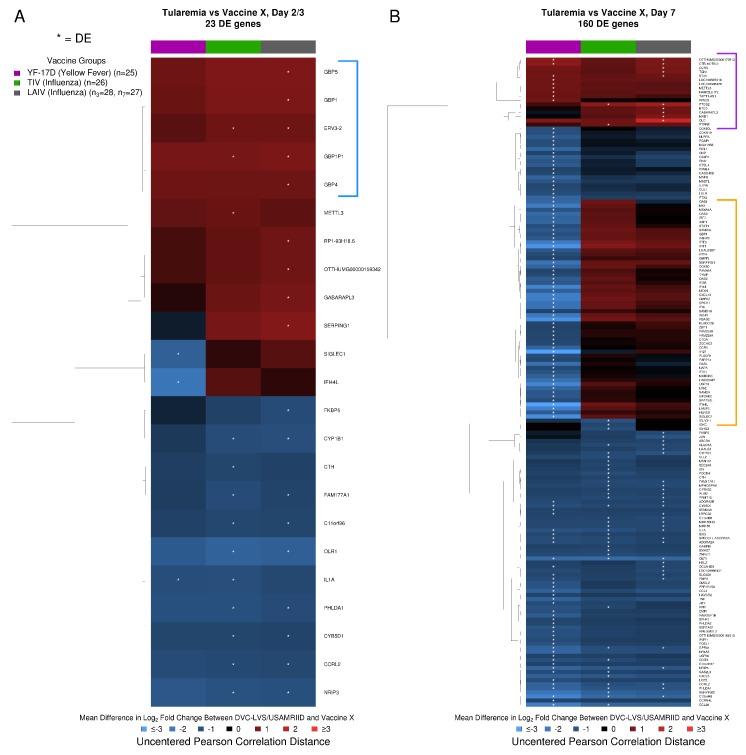
Tularemia vaccination induced unique changes in immune-related transcriptomic profiles compared to other vaccine groups. (**A**) At Day 2/3, genes encoding guanylate binding proteins (GBPs, blue square bracket; GTPases known to be IFN-γ response genes) and several other immune-related markers were significantly upregulated after tularemia vaccination relative to other vaccines. (**B**) At Day 7, 10 genes had elevated responses specifically after tularemia vaccination (purple square bracket), and 50 genes (orange square bracket) were upregulated in tularemia compared to TIV/LAIV groups but not as highly upregulated as with YF-17D. White asterisks highlight significant differentially expressed genes (DEGs) in the tularemia group compared to the other vaccines based on a two-sided Welch’s *t*-test.

**Figure 5 vaccines-08-00004-f005:**
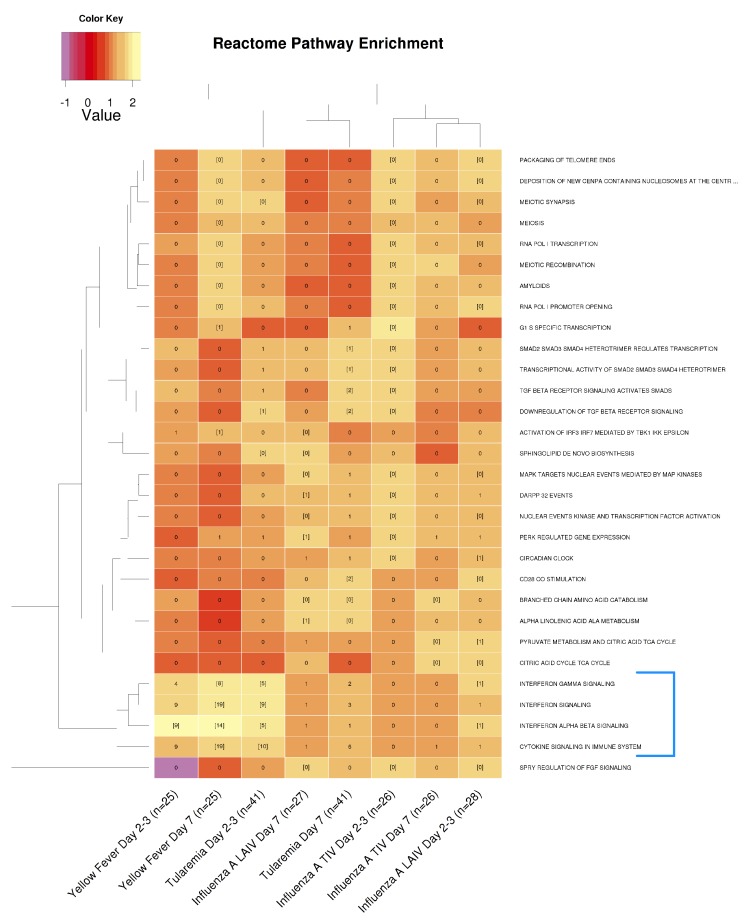
Interferon α/β signaling was more strongly induced in live, attenuated vaccine groups. Reactome pathways that were significantly enriched in at least two conditions are shown. The number of DEGs overlapping with a certain pathway is printed in each cell. Cells with gene numbers in brackets indicate significantly enriched pathways. Cells are color-coded by normalized enrichment score (NES). Gene sets and conditions were clustered based on the Euclidean distances between NES scores. Only the live attenuated vaccines (YF-17D, LAIV, and tularemia) significantly induced INF-α/β signaling-related responses at Day 2/3 (highlighted within the blue square bracket).

**Figure 6 vaccines-08-00004-f006:**
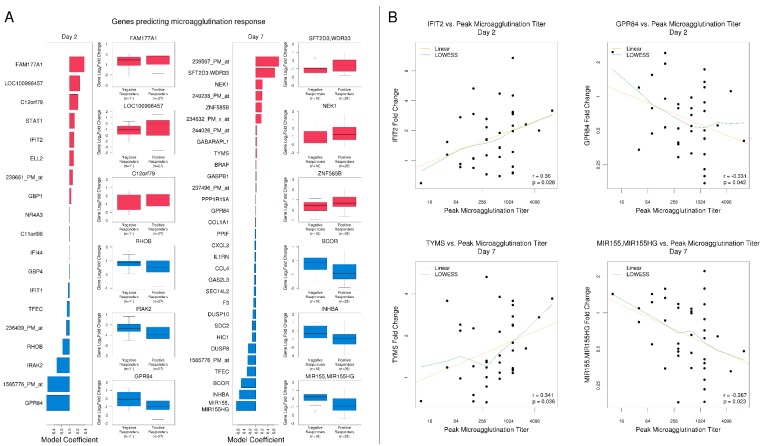
Select immune-related gene responses at Day 2 and Day 7 predicted tularemia-specific microagglutination responses. (**A**) Subjects’ microagglutination titers at Day 14 and 28 were compared to gene expression at Day 2 and 7. The bar plots represent regularized linear regression coefficients of the best model. Boxplots for the top (red) and bottom-ranked genes (blue) with annotations based on the logistic regression coefficient are presented. At Day 2, 27 subjects with positive antibody titers and 11 with negative antibody titers correlated with 19 selected gene variables. At Day 7, 28 subjects with positive titers and 10 subjects with negative titers correlated with 31 selected gene variables. (**B**) Scatterplots that summarize individual gene associations between peak microagglutination titer and gene log_2_ fold change (green lines) including locally weighted scatterplot smoothing trend lines (yellow lines) are shown for the genes that achieved the highest positive or negative Pearson correlation at Day 2 and Day 7.

**Figure 7 vaccines-08-00004-f007:**
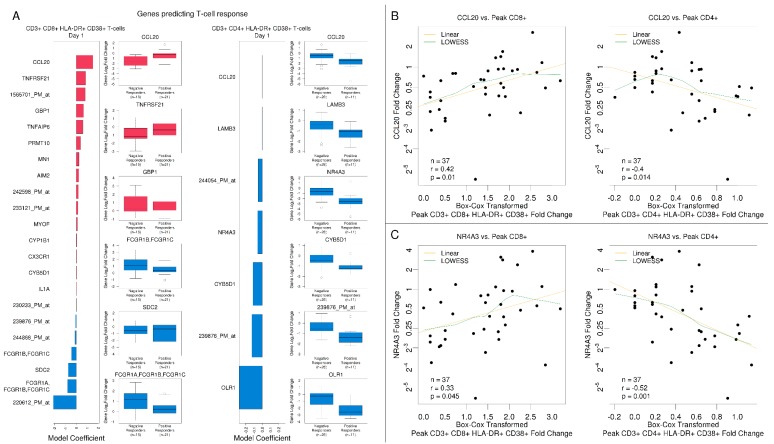
Select immune-related gene responses at Day 1 predicted T-cell responses to the tularemia vaccine. (**A**) The bar plots represent regularized linear regression coefficients of the best model for predicting a positive CD8+ response (to the left) and positive CD4+ T cell response (to the right). Boxplots for the top (red) and bottom-ranked genes (blue) with annotations based on the logistic regression coefficient are presented. (**B**,**C**) Scatterplots that summarize individual gene associations between peak CD4+ and CD8+ and gene log_2_ fold change for the CCL20 and NR4A3 genes (green lines) including locally weighted scatterplot smoothing trend lines (yellow lines). CCL20 and NR4A3 were notably negatively correlated with CD4+ T cell responses and positively correlated with CD8+ responses.

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
