# Peer review of "Systems Vaccinology for a Live Attenuated Tularemia Vaccine Reveals Unique Transcriptional Signatures That Predict Humoral and Cellular Immune Responses"

_vaccines, 2019, doi:10.3390/vaccines8010004_

Round 1

Reviewer 1 Report

A very well written manuscript that provides a better understanding of the underlying immune mechanisms involved in the development of vaccine responses to Tularemia. The manuscript compares responses to 2 unique live attenuated tularemia vaccine lots, one produced under GMP and the other that was not. These responses are further compared to a Yellow fever vaccine (YF-17D) and two influenza vaccines (TIV and LAIV).

Minor criticisms:

1. Since you identify potential differences between the DVC-LVS and USAMRIID-LVS lots it would be helpful to further understand why these two vaccine lots differ. Since the DVC-LVS is a GMP product, one might hypothesize that the differences in the development and handling of the vaccines may have contributed to the changes in response, especially in regards to the increased monocyte response observed in the USAMRIID-LVS lot. Identifying potential differences through HPLC or other biochemical means would be helpful to further understand these differences. Alternatively, expansion of the topic in the discussion would be helpful.

2. Although the authors note differences in responses to live versus inactivated vaccines, it would be useful to compare responses to Tularemia with other bacterial vaccines to determine whether differences observed in the current study are  specific for Tularemia or are just general differences that one might observe when comparing bacterial and viral vaccines. Understanding that this data may or may not be available, a discussion of this point might be appropriate.

Author Response

We would like to thank the reviewer for their kind and helpful input for our manuscript. We agree that these revisions have helped improve the content and conclusions from our work. Please see a point-by-point response below: 

Since you identify potential differences between the DVC-LVS and USAMRIID-LVS lots it would be helpful to further understand why these two vaccine lots differ. Since the DVC-LVS is a GMP product, one might hypothesize that the differences in the development and handling of the vaccines may have contributed to the changes in response, especially in regards to the increased monocyte response observed in the USAMRIID-LVS lot. Identifying potential differences through HPLC or other biochemical means would be helpful to further understand these differences. Alternatively, expansion of the topic in the discussion would be helpful.
A new sentence has been added about this point in the discussion (line 320), in addition to the other points made on this topic (line 317-319). Although the authors note differences in responses to live versus inactivated vaccines, it would be useful to compare responses to Tularemia with other bacterial vaccines to determine whether differences observed in the current study are  specific for Tularemia or are just general differences that one might observe when comparing bacterial and viral vaccines. Understanding that this data may or may not be available, a discussion of this point might be appropriate.
A new sentence has been added about future studies addressing this point (line 378).

Reviewer 2 Report

In this study, the authors employed systems vaccinology to characterize the innate and adaptive responses in human subjects with tularemia vaccination, seeking for the potential transcriptional signatures that can predict human immune responses to tularemia vaccine. Transcriptional signatures have been found as predictive biomarkers for certain viral vaccines, thus the present study using a similar approach to study tularemia vaccine is relevant and interesting. The study also compared two vaccine lots, demonstrating similar transcriptomic responses with a few notable differences. Overall, the study is well designed, and provides extent amount of data that will benefit the field of tularemia vaccine. The strength of this study is the comprehensive assessment of immune responses including immune cell populations, cytokine profile, gene expression in PBMCs and antibody responses, in a well-designed vaccination cohort. The weakness is lack of statistical analysis in immune responses comparison.

Specific comments

1. Figure 1 lacks statistical analysis. It would be helpful to include statistics to compare the immune responses between different time points, as well as between two vaccine lots. The radar plot does not present variance of the measurement. It would be helpful to specify the values (mean with SD or median with interquartile range) for those mentioned in the text.

2. The conclusion of abstract need to be revised to correspond to the title.

3. The description on study cohort is an important part. It can be moved into the main text. It’s stated 8 subjects in the main text, but 42 subjects in the study design in the supplement.

4. The conclusions part is the duplicate of the previous paragraph.

Author Response

We would like to thank the reviewer for their kind and helpful input for our manuscript. We agree that these revisions have helped improve the content and conclusions from our work. Please see a point-by-point response below: 

Figure 1 lacks statistical analysis. It would be helpful to include statistics to compare the immune responses between different time points, as well as between two vaccine lots. The radar plot does not present variance of the measurement. It would be helpful to specify the values (mean with SD or median with interquartile range) for those mentioned in the text. We have added additional tables in the supplemental materials with descriptive statistics for all of these results, lines 181-183 (Supplementary Tables 21-93). We have also updated Figure 1 and the figure legend to include statistically significant results (updated Figure 1 submitted with the revision). The conclusion of abstract need to be revised to correspond to the title. The conclusion in the abstract has been updated to include the systems vaccinology approach (line 41) The description on study cohort is an important part. It can be moved into the main text. It’s stated 8 subjects in the main text, but 42 subjects in the study design in the supplement. This number has been updated in the main text Methods Section under (line 98). In the discussion, the 8 subjects mentioned (line 369) were a small subset that we ran RNASeq on as well. The conclusions part is the duplicate of the previous paragraph. The duplicate paragraph has been removed from the discussion section and only included as the conclusions.